**Article** https://doi.org/10.1038/s41467-024-46514-2

# Tuning of the flat band and its impact on superconductivity in $Mo_5Si_{3-x}P_x$

**Rustem Khasanov** [1] ✉, **Bin-Bin Ruan** [2] ✉, **Yun-Qing Shi**[2,3], **Gen-Fu Chen** [2,3], **Hubertus Luetkens** [1], **Zhi-An Ren** [2,3] & **Zurab Guguchia** [1]

The superconductivity in systems containing dispersionless (flat) bands is seemingly paradoxical, as traditional Bardeen-Cooper-Schrieffer theory requires an infinite enhancement of the carrier masses. However, the combination of flat and steep (dispersive) bands within the multiple band scenario might boost superconducting responses, potentially explaining high-temperature superconductivity in cuprates and metal hydrides. Here, we report on the magnetic penetration depths, the upper critical field, and the specific heat measurements, together with the first-principles calculations for the $Mo_5Si_{3-x}P_x$ superconducting family. The band structure features a flat band that gradually approaches the Fermi level as a function of phosphorus doping $x$, reaching the Fermi level at $x \simeq 1.3$. This leads to an abrupt change in nearly all superconducting quantities. The superfluid density data placed on the 'Uemura plot' results in two separated branches, thus indicating that the emergence of a flat band enhances correlations between conducting electrons.

Electrons with a narrow energy dispersion in the vicinity of the Fermi level are expected to demonstrate a broad variety of physics phenomena. They form 'quasi-flat' bands, where the many-body effects are dominated over the kinetic energy. A famous example of flat band physics is the fractional quantum Hall effect, where the Hall conductance of 2D electrons shows precisely quantized plateaus at fractional values of $e^2/h$ ($e$ is the elemental charge and $h$ is the Planck constant). The quasiparticle excitations appear under a fractional filling of an electronic flat band developing in the presence of a large magnetic field[1–4]. Another example is the twisted bilayer graphene, where flat bands are formed already at zero magnetic fields[5]. There, flat electronic bands create large, many nanometre-size moiré unit cells, which results in the folding and flattening of the initial band structure of the material[6]. Such flattening plays a crucial role in the physics of bilayer graphene and leads to the appearance of strong coupling superconductivity, with the phase diagram resembling that of the high-temperature cuprates[5,7].

Both above-mentioned examples correspond to a rare case, where the bands at the Fermi level stay nearly flat. In reality, the situation with a 'quasi-flat' band is more often realized. However, this is not as rare as

one might think. The recent careful search performed by Regnault et al.[8], where more than 55,000 compounds were analyzed, presented a catalog of the naturally occurring three-dimensional stoichiometric materials with quasi-flat bands around the Fermi level. It was found, in particular, that more than 5% of all searched materials host flat band structures.

In relation to the superconducting materials, the importance of flat bands stems from the substantial decrease of the Fermi velocity, which may even tend to zero in a true flat band case. Within the conventional Bardeen–Cooper–Schrieffer (BCS) approach, this leads to a vanishingly small coherence length and the superfluid density, as well as to the extreme heavy and nearly immobile supercarriers. From the theory side, however, the emergence of flat bands is favorable to superconductivity by giving rise to a linear dependence of the transition temperature on the strength of the attractive interactions[9–12]. More interesting, the coexistence of flat and dispersive bands within the multi-band scenario leads to a strong enhancement of the transition temperature and might potentially explain the phenomena of high-temperature superconductivity in the cuprates as well as in the recently discovered hydride superconductors[13].

[1]Laboratory for Muon Spin Spectroscopy, Paul Scherrer Institute, CH-5232 Villigen PSI, Switzerland. [2]Institute of Physics and Beijing National Laboratory for Condensed Matter Physics, Chinese Academy of Sciences, 100190 Beijing, China. [3]School of Physical Sciences, University of Chinese Academy of Sciences, 100049 Beijing, China. ✉e-mail: rustem.khasanov@psi.ch; bbruan@mail.ustc.edu.cn

In this work, we probe the effect of the emergence of a flat band on the properties of the $Mo_5Si_{3-x}P_x$ superconducting family. Previous studies of $Mo_5Si_{3-x}P_x$ demonstrate the existence of superconductivity in P-doped members down to at least $x = 0.5$[14]. The appearance of a flat band right at the Fermi level leads to an abrupt change of nearly all superconducting quantities including the transition temperature $T_c$, the upper critical field $H_{c2}$, the magnetic penetration depth $\lambda$, the coherence length $\xi$, and the superconducting energy gap $\Delta$.

## Results

Figure 1a demonstrates the crystal structure of $Mo_5Si_{3-x}P_x$, refined in x-ray experiments. The system is purely three-dimensional and it has the tetragonal symmetry described by the space group $I4/mcm$ (see sec. I in the Supplemental part). The Rietveld refinements of the x-ray measurements stay in agreement with the previously published data[14]. The resistivity curves, measured at zero applied fields ($\mu_0 H_{ap} = 0.0$ T), represent metallic behavior, with the residual resistivity ratio (RRR) decreasing as the phosphorus content $x$ increases (see Fig. 1b, c). The resistance as a function of temperature changes smoothly, so there are no visible features which might be associated with some sorts of competing states and/or structural transitions.

The results of the band structure calculations of $Mo_5Si_{3-x}P_x$ are summarized in Fig. 1d. The shape of the first Brillouin zone and positions of the high symmetry points ($\Gamma$, X, M, N, and P) are shown at the top left part of Fig. 1d. The major feature of the electronic structure is the band denoted by the red color, which has a substantial flattened portion. At zero doping ($x = 0$), the flat dispersion sets at energy $\simeq$ 0.25 eV above the Fermi energy ($E_F$). With the increase in phosphorus content $x$, $E_F$ shifts to higher energies, and at $x \simeq 1.3$, the flat band dispersion approaches the Fermi level.

The doping evolution of the upper critical field $H_{c2}$ was studied in resistivity experiments. Figure 2a, b shows the resistivity curves normalized to the values at $T = 16$ K [$R(T)/R(16$ K)] measured in magnetic fields ranging from 0.0 to 9.0 T for two representative phosphorus dopings $x = 1.2$ and $x = 1.4$, respectively [the $R(T)$ curves for other doping levels are presented in the sec. II in the Supplemental part]. Obviously, the two closely doped samples, which have nearly similar superconducting transition temperatures at zero applied field, react differently on the magnetic field. As an example, at $\mu_0 H_{ap} = 9.0$ T the $x = 1.2$ sample stays in a normal state down to $T \simeq 1.75$ K, while the $x = 1.4$ one superconducts below $\simeq 5$ K.

The upper critical field values defined from $R(T)$ measurement curves are summarized in Fig. 2c. Here, $T_c(H_{ap})$'s were determined from the midpoint of $R(T, H_{ap})$ curves [i.e., as the value where $R(T)/$

$R(16$ K$) = 0.5$]. The solid lines correspond to the fits of the Ginzburg–Landau model

$$H_{c2}(T) = H_{c2}(0) \frac{1 - (T/T_c)^2}{1 + (T/T_c)^2} \quad (1)$$

to the experimental $H_{c2}(T)$ data. The values of $T_c$'s at $H_{ap} = 0$ and $H_{c2}(0)$ obtained from the fits of Eq. (1) are presented in Fig. 2d, e as a function of phosphorus doping $x$. An abrupt change of both parameters at $x \simeq 1.3$ is clearly visible.

The temperature dependencies of the magnetic penetration depth $\lambda$ were studied in transverse-field (TF) muon-spin rotation/relaxation ($\mu$SR) experiments. Measurements were performed in the field-cooling mode at the applied field $\mu_0 H_{ap} = 50$ mT. Representative TF-$\mu$SR time-spectra of $x = 1.2$ and 1.4 samples at $T \simeq 1.5$ K (i.e., below $T_c$) are shown in Fig. 3a, b. A strong damping reflects the inhomogeneous field distribution $P(B)$ caused by the formation of the flux-line lattice (FLL). The broad asymmetric distribution is clearly visible in Fig. 3c, d, where the Fourier transforms of the corresponding TF-$\mu$SR time-spectra are shown. The $P(B)$ distributions demonstrate all characteristic features of a well-arranged FLL, namely the cut-off at low-field, the extended tail to the higher field values, and the shift of the $P(B)$ peak below $\mu_0 H_{ap}$[15]. Note that the narrow peak at the applied field position ($B_{ap} = \mu_0 H_{ap}$) originates from muons missing the sample.

The width of $P(B)$ within the FLL in the limit of $H_{ap} \ll H_{c2}$ [as is the case for our studies, see Fig. 2c, e] is primarily determined by the value of the magnetic penetration depth $\lambda$[15-17]. Comparison of $P(B)$'s presented in Fig. 3c, d suggests that the $x = 1.2$ sample has stronger broadening (i.e., smaller $\lambda$ value) compared to that of the $x = 1.4$ one.

The temperature dependencies of the inverse squared magnetic penetration depth of $Mo_5Si_{3-x}P_x$ samples are presented in Fig. 3e. For all dopings $\lambda^{-2}(T)$ demonstrates saturation for $T \lesssim 4$ K (i.e., for temperature below ~1/3 of $T_c$), which is consistent with the formation of a fully gapped state. The solid lines represent the best fits within the s-wave BCS model[18]

$$\frac{\lambda^{-2}(T)}{\lambda^{-2}(0)} = 1 + 2 \int_{\Delta(T)}^{\infty} \left( \frac{\partial f}{\partial E} \right) \frac{E \, dE}{\sqrt{E^2 - \Delta(T)^2}} . \quad (2)$$

Here $f = [1 + \exp(E/k_B T)]^{-1}$ is the Fermi function and $\Delta(T) = \Delta(0) \tanh\{1.82[1.018(T_c/T - 1)]^{0.51}\}$ is the temperature dependent superconducting gap[19]. $\lambda^{-2}(0)$ and $\Delta(0)$ are the zero-temperature values of the inverse squared penetration depth and the

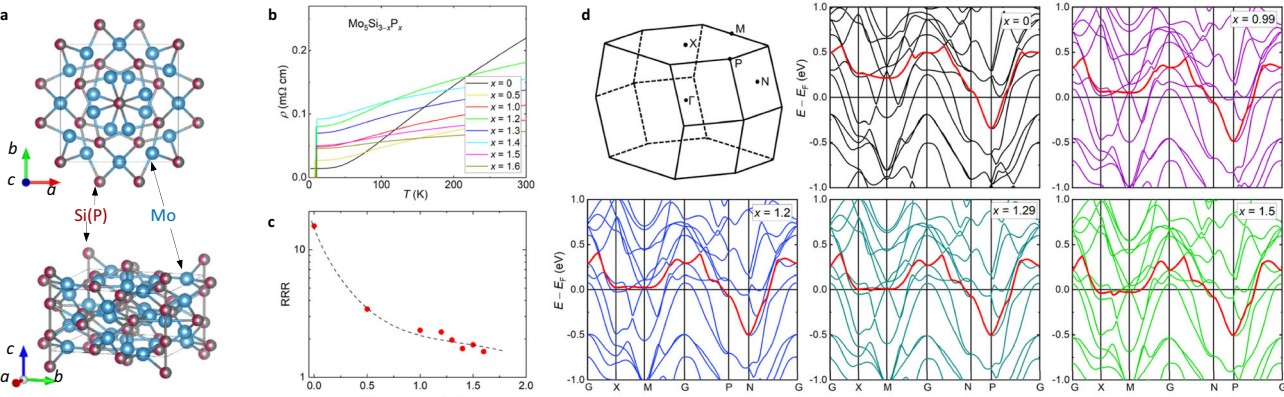

**Fig. 1 | The crystal structure, the residual resistivity, and the electronic band structure of $Mo_5Si_{3-x}P_3$. a** The crystal structure of $Mo_5Si_{3-x}P_x$. **b** The resistivity curves of $Mo_5Si_{3-x}P_x$ ($0.0 \leq x \leq 1.6$) measured at a zero-applied field. The superconducting transition in $Mo_5Si_{3-x}P_x$ is detected for $x \gtrsim 0.5$[14]. **c** Evolution of the residual resistivity ratio (RRR) as a function of phosphorus content $x$. **d** The electronic band structures of $Mo_5Si_{3-x}P_x$. The top left panel represents the first Brillouin zone of $Mo_5Si_{3-x}P_x$ with high-symmetry points labeled as $\Gamma$, X, M, N, and P. The band displaying an extended flattened portion is denoted by the red color.

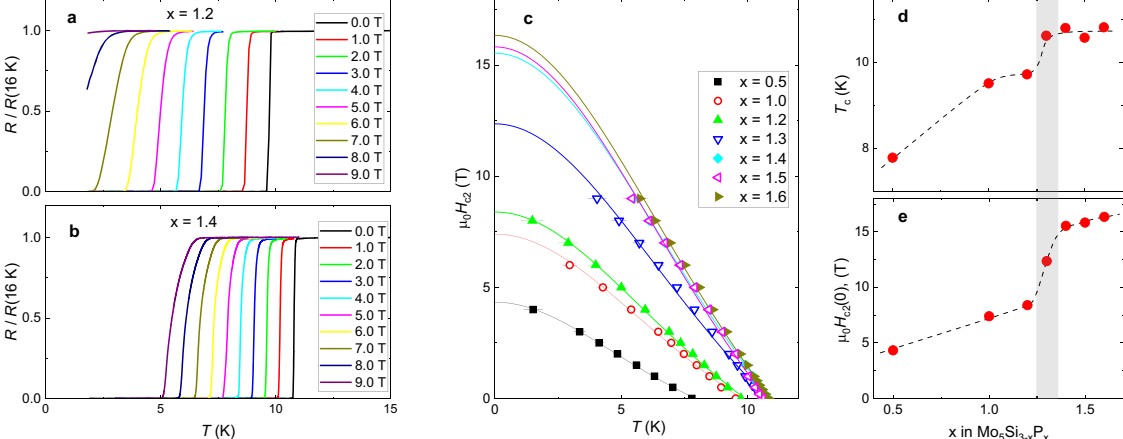

**Fig. 2 | Doping dependencies of the superconducting transition temperature and the upper critical field. a** and **b** Temperature dependencies of resistivity of $Mo_5Si_{3-x}P_x$ ($x = 1.2$ and $1.4$) measured in magnetic fields ranging from 0.0 to 9.0 T. **c** Temperature dependencies of the upper critical field $H_{c2}$. The lines are fits of Eq. (1) to the data. **d** Doping dependence of the transition temperature $T_c$ at $H_{ap} = 0$. **e** Doping dependence of the zero-temperature value of the upper critical field $H_{c2}(0)$. The lines in d and e are guides for the eye. The gray stripe represents the region at $x \simeq 1.3$, corresponding to an abrupt change of $T_c$ and $H_{c2}(0)$. The error bars for $H_{c2}(T)$ indicate uncertainty in selecting the midpoint of $R(T, H_{ap})$ curves. The error bars for $H_{c2}(0)$ and $T_c$ data correspond to one standard deviation from the $\chi^2$ fit of $H_c(T)$ by means of Eq. (1).

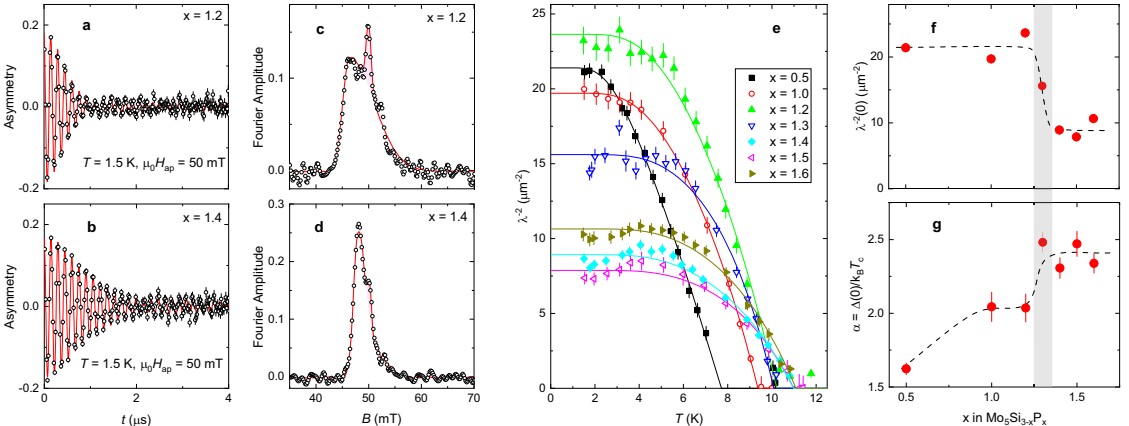

**Fig. 3 | Doping dependencies of the superfluid density and the gap to $T_c$ ratio.** **a** and **b** The muon-time spectra of $Mo_5Si_{3-x}P_x$ ($x = 1.2$ and $1.4$) measured at $T = 1.5$ K and $\mu_0H_{ap} = 50$ mT. The solid lines are fits of Eq. (5) to the data. **c** and **d** The Fourier transforms of asymmetry spectra are presented in panels (**a** and **b**). Red peaks denote the background contribution originating from muons missing the sample. **e** Temperature evolutions of $\lambda^{-2}$. The solid lines are fits of Eq. (2) to the data. **f** Doping dependence of $\lambda^{-2}(0)$. **g** Doping dependence of the gap to $T_c$ ratio $\alpha = \Delta(0)/k_BT_c$. The lines in f and g are guides for the eye. The gray stripe represents the region at $x \simeq 1.3$, corresponding to an abrupt change of $\lambda^{-2}(0)$ and $\alpha$. The displayed error bars for parameters obtained from $\mu$SR data correspond to one standard deviation from the $\chi^2$ fits.

superconducting gap, respectively. The dependencies of the fit parameters, namely $\lambda^{-2}(0)$ and $\alpha = \Delta(0)/k_BT_c$ on the phosphorus content $x$ are summarized in Fig. 3f, g, respectively. A step-like change of both parameters takes place at $x \simeq 1.3$.

## Discussion

The results obtained in resistivity (Fig. 2) and TF-$\mu$SR (Fig. 3) experiments imply that the major superconducting quantities, namely the transition temperature $T_c$, the upper critical field $H_{c2}$, the magnetic penetration depth $\lambda$, and the energy gap $\Delta(0)$ demonstrate an abrupt change at $x \simeq 1.3$. Two possible scenarios can be considered. The first one assumes the formation of a competing ordered state, where part of the carriers are gapped due to competing interactions and, therefore, becomes inaccessible for the Cooper pair formation. As an example of such states, one may refer to the charge-density-wave (CDW) or spin-density wave (SDW) type of orders, which are widely detected for cuprate, kagome, and Fe-based superconducting families[20-28]. This scenario is not plausible here, since (i) The resistivity

experiments presented in Fig. 1b do not detect any features at the normal-state resistivity curves up to $T \simeq 300$ K. (ii) The specific heat experiments reveal the absence of an abrupt change of the density of states at the Fermi level [$N(E_F)$] in the vicinity of $x \simeq 1.3$ (see secs. III and IV in the Supplemental Part). (iii) The zero-field $\mu$SR experiments do not detect any kind of magnetism (see sec. VI in the Supplemental Part), thus implying that the SDW type of order does not come into play.

The second scenario assumes the emergence of a flat band at the Fermi level, in analogy with that discussed for kagome superconductor $LaRu_3Si_2$[29]. The results of band-structure calculations presented in Fig. 1d demonstrate the presence of a substantial flattened band portion (denoted by the red color) and indicate that above the critical doping level ($x \gtrsim 1.3$) the flat band comes into play. It is remarkable that the doping level, at which the flat band approaches $E_F$, coincides with the level where all the measured superconducting quantities [$T_c$, $H_{c2}(0)$, $\lambda^{-2}(0)$, and $\Delta(0)$] demonstrate abrupt changes (see Figs. 2d, e and 3f and g).

The effects of band flattening on the two fundamental superconducting length scales, namely the magnetic penetration depth $\lambda$ (which defines a distance for magnetic field decay) and the coherence length $\xi$ (which determines the size of a Cooper pair), might be understood in relation to the corresponding changes of the Fermi velocity $v_F$. Note that all these quantities are obtainable from the above-presented data: the value of $\xi$ could be calculated from the measured $H_{c2}$ by using the Ginzburg–Landau expression $H_{c2} = \Phi_0/2\pi\xi^2$ [30], $\lambda$ is measured directly in TF-$\mu$SR experiments and the Fermi velocity $v_F$ might be estimated from the electronic structure as the first derivative of the band dispersions at $E_F$.

For a single-band superconductor and within the conventional BCS scenario, the zero-temperature values of the coherence length and the penetration depth follow the well-known relations:

$$\xi(0) = \frac{\hbar\langle \mathbf{v}_F\rangle}{\pi\Delta(0)} = \frac{1}{\pi\alpha}\frac{\hbar\langle \mathbf{v}_F\rangle}{k_B T_c} \qquad (3)$$

and

$$\lambda(0) = \sqrt{\frac{m^*}{\mu_0 n_s e^2}} = \sqrt{\frac{\hbar\langle \mathbf{k}_F\rangle}{\langle \mathbf{v}_F\rangle n_s e^2}} \qquad (4)$$

Here $\hbar$ is the reduced Planck constant, $\langle \mathbf{v}_F\rangle$ is the average value of the Fermi velocity, $n_s$ is the charge carrier concentration, $m^* = \hbar\langle \mathbf{k}_F\rangle/\langle \mathbf{v}_F\rangle$ is the effective carrier mass, and $\langle \mathbf{k}_F\rangle$ is the averaged Fermi wave vector. By having only limited validity for Mo$_5$Si$_{3-x}$P$_x$, which is definitively not a single-band, but a multi-band superconductor (see Fig. 1d), the above equations can still capture the main features of our experimental observation: (i) The flat band approaches the Fermi level at $x \gtrsim 1.3$, which leads to a sudden decrease of $\langle \mathbf{v}_F\rangle$. Following Eqs. (3) and (4), this requires the coherence length $\xi(0)$ to decrease, and $\lambda(0)$ to increase accordingly. The corresponding effects on the measured quantities $H_{c2}(0)$ and $\lambda^{-2}(0)$ are just the opposite of those for $\xi(0)$ and $\lambda(0)$, in agreement with the experimental observations (Figs. 2e and 3f). (ii) A contrasting band flattening effect on $\xi(0)$ and $\lambda(0)$ would imply a strong change of the Ginzburg-Landau parameter $\kappa = \lambda/\xi$. This is demonstrated in Fig. 4a, where $\kappa(0)$ increases by nearly three times from $\simeq 25$ for $x < 1.3$ to $\simeq 70$ for $x > 1.3$.

Additional information could be obtained by comparing the $T_c$ vs. $\lambda^{-2}(0)$ dependence for Mo$_5$Si$_{3-x}$P$_x$ with the so-called 'Uemura plot' for various classes of unconventional superconducting materials, Fig. 4b[31-36]. The solid, dashed, and dash-dotted lines correspond to the 'Uemura plots' for the hole-doped and the electron-doped high-

temperature cuprate superconductors[31-33], and for the transition metal dechalcogenides (TMDs)[34-36], respectively. The ratio $T_c/\lambda^{-2}(0)$ for these three classes of unconventional superconductors is significantly larger than that of conventional BCS superconductors, indicative of a much smaller superfluid density. It is widely discussed that the difference between $T_c/\lambda^{-2}(0)$ ratios (which appear to be nearly constant within particular classes of unconventional superconductors) may become a measure of the effects of strong correlations between the conducting electrons. The data presented in Fig. 4b show that above and below the critical doping $x \simeq 1.3$ the experimental points follow the tendency for TMDs and the electron-doped cuprate superconductors, respectively. This implies that the emergence of the flat band in Mo$_5$Si$_{3-x}$P$_x$ enhances the effects of electronic correlations.

To conclude, measurements of the magnetic penetration depths, the upper critical field, and the specific heat, together with the first-principles calculations, in the newly discovered superconductor family Mo$_5$Si$_{3-x}$P$_x$ were carried out. In accordance with previous studies, the superconductivity in Mo$_5$Si$_{3-x}$P$_x$ exists down to $x \cdot 0.5$[14]. For the phosphorus content $x$ exceeding $\simeq 1.3$, the calculated band structure features a flat band right at the Fermi level, followed by an abrupt change of nearly all superconducting quantities. In particular, the transition temperature $T_c$ increases by $\simeq 15\%$ (from $\simeq 9.5$ to nearly 11 K), the upper critical field $H_{c2}$ increases by more than a factor of two (from $\simeq 7$ to $\simeq 16$ T), and the inverse squared magnetic penetration depth (which is normally considered to be a measure of the supercarrier concentration) decreases by more than twice (from $\simeq 22$ to $\simeq 9$ $\mu$m$^{-2}$). Our results point to the unprecedented case of bulk system Mo$_5$Si$_{3-x}$P$_x$ in which superconducting properties can be manipulated by controlling the location of flat band with respect to the Fermi level and offer unique insights into the role played by band flattening in the superconducting mechanism. Engineering through strain and pressure allows for the manipulation of phases in a controlled manner, potentially facilitating further exploration of the superconducting phase.

## Methods
### Sample preparation
Polycrystalline Mo$_5$Si$_{3-x}$P$_x$ samples with $x = 0.5, 1.0, 1.2, 1.3, 1.4, 1.5$, and 1.6 were prepared by a solid-state reaction[14]. The procedure included initially mixing and pressing elemental powders of Mo (99.9% purity), Si (99.999% purity), and P (99.99% purity) into pellets, followed by two subsequent annealings: first at 1073 K for 24 h and second at 1923 K for 20 h.

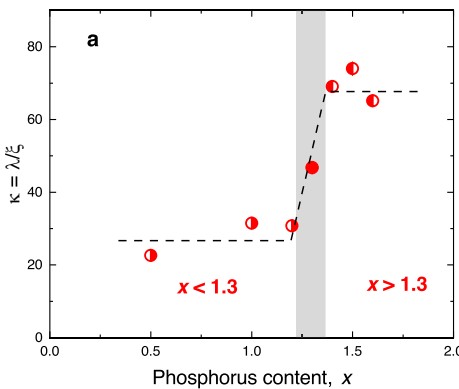

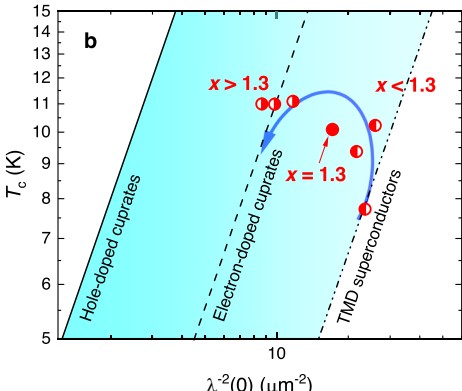

**Fig. 4 | Doping evolution of the Ginzburg–Landau parameter and the 'Uemura plot'. a** The dependence of the Ginzburg–Landau parameter $\kappa = \lambda(0)/\xi(0)$ of Mo$_5$Si$_{3-x}$P$_x$ on the phosphorus content $x$. At the critical doping, $x \simeq 1.3$, a transition from the steep band ($x < 1.3$) to the steep band/flat band ($x > 1.3$) type of behavior occurs. **b** The 'Uemura plot' for various families of unconventional

superconductors[31-36]. Above and below the critical doping $x \simeq 1.3$, the experimental $\lambda^{-2}(0)$ vs. $T_c$ points for Mo$_5$Si$_{3-x}$P$_x$ follow the tendency for TMD and the electron-doped cuprate superconductors, respectively. The error bars for $\kappa$ are defined from those for $\lambda(0)$ and $\xi(0)$.

The crystal structure and phase purity were checked by powder x-ray diffraction, confirming the tetragonal structure (space group $I4/mcm$). In addition to the main $Mo_5Si_{3-x}P_x$ phase, a small amount of impurity phase $Mo_3P$ (from $\simeq$5% to 10%) was detected (see sec. I in the Supplemental Part).

### X-ray, resistivity, and specific heat experiments

The room-temperature powder x-ray diffraction (XRD) data were collected on a PAN-analytical X-ray diffractometer with Cu-K$\alpha$ radiation. Rietveld refinements were carried out using the GSAS package[37]. The resistivity and heat capacity data were collected on a physical property measurement system (PPMS, Quantum Design).

### First principle calculations

The first-principles calculations were performed based on the density functional theory, as implemented in the Quantum ESPRESSO package[38]. The generalized gradient approximation of Perdew–Burke–Ernzerhof (PBE) exchange-correlation functionals was applied[39]. The optimized norm-conserving pseudopotentials were chosen[40]. The energy cutoffs for wavefunction and charge densities were 50 and 400 Ry, respectively. Before each calculation, the cell dimensions and atomic positions were fully relaxed until each atom felt a force of $<1\times10^{-4}$ Ry Bohr$^{-1}$. Phosphorous doping was treated by the virtual crystal approximation (VCA), whose validation had been checked in ref. 14. Monkhorst–Pack grids of $11\times11\times7$ and $21\times21\times15$ were used to calculate the charge densities and the density of states (DOS), respectively. No Hubbard parameters or spin–orbit coupling effects were taken into account, as they introduced no obvious change to the electronic band structure and DOS, as demonstrated in ref. 14.

### Muon-spin rotation/relaxation experiments

The muon-spin rotation/relaxation ($\mu$SR) measurements were carried out at the $\pi$M3 beamline using the general purpose surface (GPS) $\mu$SR spectrometer (Paul Scherrer Institute, Villigen, Switzerland)[41]. In this study, we primarily performed transverse-field (TF) $\mu$SR measurements, which allowed us to determine the temperature evolution of the magnetic penetration depth. The $\mu$SR data were analyzed by means of the Musrfit software package[42].

### $\mu$SR data analysis procedure

The analysis of TF-$\mu$SR data was performed by considering the presence of a main $Mo_5Si_{3-x}P_x$ phase (denoted as s) and two background contributions (bg,1 and bg,2), respectively. The bg,1 contribution originates from the impurity $Mo_3P$ phase (which superconducts at $T_c \simeq 5.5$ K, refs. 43–45), while the bg,2 one is caused by muons missing the sample (i.e., stopped at the sample holder and the cryostat windows). The following functional form was used:

$$A(t) = A_s \, \mathrm{SkG}(t, B_s, \sigma_+, \sigma_-) + A_{\mathrm{bg},1} \, e^{-\sigma_{\mathrm{bg},1}^2 t^2/2}$$
$$\times \cos(\gamma_\mu B_{\mathrm{bg},1} + \phi) + A_{\mathrm{bg},2} \cos(\gamma_\mu B_{\mathrm{ap}} + \phi). \tag{5}$$

Here $A_s$ (~90%), $A_{\mathrm{bg},1}$ (~10%), and $A_{\mathrm{bg},2}$ (~1%) are the initial asymmetries, and $B_s$, $B_{\mathrm{bg},1}$, and $B_{\mathrm{ap}}$ are the internal fields of each particular component. $\gamma_\mu = 2\pi \cdot 135.53$ MHz/T is the muon gyromagnetic ratio, $\phi$ is the initial phase of the muon–spin ensemble, and $\sigma$ is the Gaussian relaxation rate. The sample contribution was fitted with the Skewed Gaussian function [$\mathrm{SkG}(B, \sigma_+, \sigma_-)$], which accounts for the asymmetric $P(B)$ distribution within the FLL[46,47]. The second central moment of the sample contribution $\langle\Delta B^2\rangle_s$ was obtained from the fitted $\sigma_+$ and $\sigma_-$ values[46]. The inverse squared magnetic penetration depth was further calculated as $\lambda^{-2}[\mu m^{-2}] = 9.32 \times \sqrt{\langle\Delta B^2\rangle_s - \sigma_{\mathrm{nm}}^2} \, [\mu s^{-1}]$[17,47]. Here $\sigma_{\mathrm{nm}}$ is the nuclear moment contribution which is determined from the measurements above $T_c$[15,47].

## Data availability

All relevant data are available from the authors. The data can also be found at the following link http://musruser.psi.ch/cgi-bin/SearchDB.cgi.

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

## Acknowledgements

Z.G. acknowledges support from the Swiss National Science Foundation (SNSF) through the SNSF Starting Grant (No. TMSGI2_211750). Z.-A.R. acknowledges support from the National Key Research and Development Program of China (Grant Nos. 2018YFA0704200 and 2021YFA1401800) and the National Natural Science Foundation of China (Grant No. 12074414).

## Author contributions

R.K. conceived and supervised the project. B.-B.R. and Y.-Q.S. synthesized the samples and performed resistivity, x-ray, and specific heat experiments. R.K. performed the μSR experiments. B.-B.R. performed the band-structure calculations. R.K. wrote the manuscript with contributions from B.-B.R., Y.-Q.S, G.-F.C, H.L., Z.-A.R., and Z.G.

## Competing interests

The authors declare no competing interests.
