## [Peer Review File · Nature Communications]

Tuning of the flat band and its impact on superconductivity in $\text{Mo}_5\text{Si}_{3-x}\text{P}_x$REVIEWER COMMENTS

Reviewer #1 (Remarks to the Author):

Khasanov et al. report a combined experimental and theoretical study on the $\text{Mo}_5\text{Si}_3\text{-xPx}$ superconducting family. They found that a flat band approaches the Fermi level at x round 1.3, leading to a strong reduction of the coherence length, an enhancement of the penetration depth, and a large increase of the Ginzburg-Landau parameter. This underscores a transition from a moderate to an extreme type-II superconducting behavior, which is further corroborated by two separated branches of the superfluid density data. The overall results indicates that the emergence of a flat band leads to enhancement of electron correlations, which is of interest and presented in an organized way. However, the data analysis is inadequate and needs to be improved before the paper can be considered for publication. My specific comments are:

1) It is known that the $H_{c2}(0)/T_c$ is proportional to $\gamma\rho_N$ in the dirty limit (as in the present case), where γ is the Sommerfeld coefficient and ρ_N is the normal state resistivity just above T_c . From Fig. 1(b), one can see that the low-temperature resistivity increases significantly upon P doping. It is thus natural to expect that $H_{c2}(0)/T_c$ tends to increase with increasing P content, as observed experimentally in Fig. 2(c). Note that this explanation does not necessarily involve the emergence of a flat band near the Fermi level and hence should be examined carefully.

2) In both Fig. 2(d) and Fig. 3(g), a plateau is observed for x between 1.0 and 1.2. What could be the reason for this?

3) Is there any other experimental evidence for the approaching of a flat band towards the Fermi level? For example, how the temperature dependence of Hall coefficient evolves with P doing. This also helps to clarify the dopant nature of P.

4) The authors explained the variations of the coherence length and penetration depth as due to the change in Fermi velocity. Can the latter quantity be determined by the theoretical calculations? If so, a comparison with the experimental results should be done to better support the argument. In addition to main issues, several typos are found in the manuscript. For example, in the 2nd paragraph of page 4, "this requires the coherence length $\xi(0)$ to increase, and $\lambda(0)$ to decrease" should be changed to "this requires the coherence length $\xi(0)$ to decrease, and $\lambda(0)$ to increase".

Reviewer #2 (Remarks to the Author):

Review report on; "Emergence of flat bands and their impact on superconductivity in $\text{Mo}_5\text{Si}_3\text{-xPx}$ ", by "Khasanov" et al.

The authors report on synthesis, characterization, and analysis of a series of P-doped Mo_5Si_3 compounds through a suite of experiments alongside band-structure calculations.

The subject matter of the manuscript is topical and of current interest, and deserves to be considered for publication within the condensed-matter physics community.

The main message is the apparent facile manipulation of flat electronic bands, and which may be used as a control parameter in studying superconductivity in certain materials classes. The authors seek to position two different doping levels of the title material among electron-doped cuprate superconductors and those composed of transition metal dichalcogenides, respectively.

The experiments are of good quality, and suitably represent the physics that the authors are pursuing in this manuscript.

The manuscript is easy to follow and in most cases the arguments are plausible and supported by the data.

However, a number of issues stand in the way for me to recommend the manuscript towards publication in a journal with the audience of Nature Communications.

First, I am not convinced that the message of the manuscript bears enough novelty, nor an advance in the field, nor the substance that would justify its publication in Nature Communications.

Second, the text in the manuscript is unfortunately replete with grammar and syntax errors, and thorough language editing would be needed in case this manuscript were to receive further consideration for publication. In many cases the lack of care afforded to grammar results in the text becoming difficult to understand ("From the theory site [sic], however, the emergence of flat bands stay in a favor to superconductivity by giving rise to a linear dependence of the transition temperature...") or technically incorrect ("...the fractional quantum hole effect...").

Below, I list a number of points for the authors to consider.

1. The manuscript deals with a series of P-doped members of the parent compound Mo_5Si_3 . Previous studies by some of the present authors have demonstrated the existence of superconductivity in P-doped members down to at least $x=0.5$, but superconduction is absent in the parent (undoped) compound. I found it a shortcoming of the present manuscript that practically no information is provided on the parent compound. See also point 2 below.
2. The title of the manuscript conveys the notion that flat electronic bands emerge upon P-doping of the parent compound. While some degree of P-doping is evidently needed for the system to become superconducting (the parent compound remains in the normal state at least down to 150 mK, according to reference 38), the existence of a section of flat bands can be found even in the parent compound, judged by figure 1(d) ($x=0$ panel). Doping then evidently achieves to shift the Fermi level towards the flat band. The use of the descriptor "emergence" in the title and elsewhere thus seems questionable. Instead, "tuning of the flat band" or wording to this effect seems to be a more accurate description of the situation. This point also concerns, for example, the phrasing of observation (i) on p4.
3. p2, left-hand column below figure 1: "Figs. 1 (c) and (d)" - this should be (b) and (c).
4. Figure 1(a): missing labels for the different elements in the structure representation.
5. Figure 1, caption: "The top right panel represent the first Brillouin zone...". As stated elsewhere in the text, this should instead read "The top left panel...".
6. Figure 1(b): The colors used in the legend for fields of respectively 0 T and 9 T are the wrong way round.

end of report

Please find enclosed our responses to the criticism and suggestions of the Reviewers. The changes introduced into the manuscript are denoted in red color.

Reply to the Reviewer # 1

We are grateful to Reviewer # 1 for valuable comments.

- 1) *It is known that the $H_{c2}(0)/T_c$ is proportional to $\gamma\rho_N$ in the dirty limit (as in the present case), where γ is the Sommerfeld coefficient and ρ_N is the normal state resistivity just above T_c . From Fig. 1(b) one can see that the low-temperature resistivity increases significantly upon P doping. It is thus natural to expect that $H_{c2}(0)/T_c$ tends to increase with increasing P content, as observed experimentally in Fig. 2(c). Note that this explanation does not necessarily involve the emergence of a flat band near the Fermi level and hence should be examined carefully.*

Figure 1 shows the dependence of $\gamma \times \rho(T = 15\text{K})$ on $H_{c2}(0)/T_c$ for $\text{Mo}_5\text{Si}_{3-x}\text{P}_x$ samples studied in the present work. There is no correlation between these two quantities.

FIG. 1: The dependence of $\gamma \times \rho(T = 15\text{K})$ on $H_{c2}(0)/T_c$ for $\text{Mo}_5\text{Si}_{3-x}\text{P}_x$ samples studied our work.

2) In both Fig. 2(d) and Fig. 3(g), a plateau is observed for x between 1.0 and 1.2. What could be the reason for this?

The reason for plateau at $x \simeq 1.0 - 1.2$ might be the existence of two distinct superconducting regimes occurring above and below the critical P content $x \simeq 1.3$. In some respect, it might be similar to a ‘multidome’ structure, which is often detected for various families of unconventional superconductors as a function of pressure, doping, *etc.* The first dome ‘saturates’ [reaches the maximum $T_c(x)$ values] at around $x \sim 1.0 - 1.1$, thus demonstrating a plateau, Fig. 2s. The same might be true for the coupling constant $\alpha = \Delta(0)/k_B T_c$, Fig. 3 (g) in the main text.

FIG. 2: The dependence of the transition temperature T_c on phosphorus content x of $\text{Mo}_5\text{Si}_{3-x}\text{P}_x$. The colored areas represent two superconducting areas (‘domes’) where the purely ‘steep band’ and the ‘steep bands + flat bands’ regimes are realised.

3) Is there any other experimental evidence for the approaching of a flat band towards the Fermi level? For example, how the temperature dependence of Hall coefficient evolves with P doing. This also helps to clarify the dopant nature of P .

To trace the dopant nature of P , the series of Hall coefficient measurements were performed, Fig. 3. From the results presented in Figure 3 the following four important points emerge:

1. At room temperature, the dominant carriers in the parent Mo_3P_5 compound are holes, while in all the doped samples ($x \geq 0.5$) the electron-type carriers are dominant;
2. Phosphorus acts as an electron donor, as inferred from negative values of the Hall coefficient R_H for temperatures above $\simeq 100$ K. Note that the high-temperature region may reflect better the intrinsic doping, since above a fraction of the Debye temperature $\simeq 0.2\Theta_D - 0.4\Theta_D$, the scattering of carriers becomes isotropic [Phys. Rev. Lett., **72**, 2636 (1994)].

3. For $x \leq 1.0$, R_H is highly temperature-dependent. It even changes the sign with increasing temperature. While for $x \geq 1.2$, R_H becomes less sensitive to temperature. This phenomenon resembles that in $\text{Ba}_{1-x}\text{K}_x\text{Fe}_2\text{As}_2$, where R_H changes significantly in lower doping levels but remains almost unchanged when approaching the optimal doping level [Phys. Rev. B **85**, 064522 (2012)].
4. In the vicinity of the superconducting transition ($T \sim 10 - 15$ K), the type of dominant carriers changes from the hole-type for $x \leq 1.0$ to electron-type for $x \geq 1.3$, respectively.

Note that the points 2 and 3 are consistent with DFT calculation results [see Figure 1 in the main text and Figure 3 below].

FIG. 3: Temperature evolution of the Hall coefficient R_H of $\text{Mo}_5\text{Si}_{3-x}\text{P}_x$ ($0 \leq x \leq 1.5$).

- 4) *The authors explained the variations of the coherence length and penetration depth as due to the change in Fermi velocity. Can the latter quantity be determined by the theoretical calculations? If so, a comparison with the experimental results should be done to better support the argument.*

A new set of DFT calculations was performed. The Fermi velocities for the electron-type and the hole-type electron pockets are presented in Figure 4. By increasing x , the hole-type pockets shrink, while the electron-type pockets expand. In addition, a sixth electron-type pocket emerges for $x \geq 0.99$, thus suggesting that P doping adds electrons to the system, which is consistent with the Hall measurements.

The data presented in Fig. 4 enabled the calculation of the average Fermi velocities ($\langle v_F \rangle$ s). Figure 5 shows the theoretical $\langle v_F \rangle$ values obtained via averaging over narrow energy regions at the Fermi surfaces. Figure 5 reveals an overall decreasing trend of $\langle v_F \rangle$ with increasing P doping. It should be noted, however, that the anisotropic scattering could be significant at low temperature, limiting the validity of theoretical estimation of $\langle v_F \rangle$.

FIG. 4: Fermi velocities at various Fermi surface sheets of $\text{Mo}_5\text{Si}_{3-x}\text{P}_x$ ($0 \leq x \leq 1.62$) as obtained from DFT calculations.

Alternatively, the Fermi velocity v_F , the mean free path l_e , and the effective mass m^* might be estimated by following the approach presented in [Phys. Rev. B **96**, 064521 (2017)]. The corresponding quantities might be obtained from the following set of equations:

$$\gamma_V = \left(\frac{\pi}{3}\right)^{2/3} \frac{m^* k_B^2 n^{1/3}}{\hbar^2}, \quad (1)$$

$$l_e = \frac{3\pi^2 \hbar^3}{e^2 \rho_0 m^{*2} v_F^2}, \quad (2)$$

$$n = \frac{1}{3\pi^2} \left(\frac{m^* v_F}{\hbar}\right)^3, \quad (3)$$

where γ_V is the Sommerfeld coefficient per standard volume, n is the carrier concentration, and ρ_0 is the residual resistivity. The evolution of v_F , l_e , and m^* as a function of x are presented in Fig. 6. Figure 6 shows that the mean free path l_e stays nearly constant for $x > 0.5$, while both, v_F and m^* , obey considerable changes at $x \sim 1.3$.

- 5) *In addition to main issues, several typos are found in the manuscript. For example, in the 2nd paragraph of page 4, "this requires the coherence length $\xi(0)$ to increase, and $\lambda(0)$ to decrease" should be changed to "this requires the coherence length $\xi(0)$ to decrease, and $\lambda(0)$ to increase".*

The typos are corrected

FIG. 5: Average Fermi velocity of $\text{Mo}_5\text{Si}_{3-x}\text{P}_x$ ($0 \leq x \leq 1.62$) as obtained from DFT calculation.

FIG. 6: Dependence of the Fermi velocity v_F (right panel), the mean-free path l_e (middle panel), and the carrier mass m^* on the P content x .

Reply to the Reviewer #2

We are grateful to Reviewer #2 for useful comments. Below, we respond to Reviewer #2's criticisms point by point.

1. *The manuscript deals with a series of P-doped members of the parent compound Mo_5Si_3 . Previous studies by some of the present authors have demonstrated the existence of superconductivity in P-doped members down to at least $x = 0.5$, but superconduction is absent in the parent (undoped) compound. I found it a shortcoming*

of the present manuscript that practically no information is provided on the parent compound. See also point 2 below.

In the revised version of the manuscript, we have added sentences in the introductory, Figure 1 caption, and concluding parts as: "Previous studies of $\text{Mo}_5\text{Si}_{3-x}\text{P}_x$ demonstrate the existence of superconductivity in P-doped members down to at least $x = 0.5$.¹⁴", "The superconducting transition in $\text{Mo}_5\text{Si}_{3-x}\text{P}_x$ is detected for $x \gtrsim 0.5$.¹⁴", and "In accordance with previous studies, the superconductivity in $\text{Mo}_5\text{Si}_{3-x}\text{P}_x$ exists down to $x \sim 0.5$.¹⁴".

- 2. The title of the manuscript conveys the notion that flat electronic bands emerge upon P-doping of the parent compound. While some degree of P-doping is evidently needed for the system to become superconducting (the parent compound remains in the normal state at least down to 150 mK, according to reference 38), the existence of a section of flat bands can be found even in the parent compound, judged by figure 1(d) ($x=0$ panel). Doping then evidently achieves to shift the Fermi level towards the flat band. The use of the descriptor "emergence" in the title and elsewhere thus seems questionable. Instead, "tuning of the flat band" or wording to this effect seems to be a more accurate description of the situation. This point also concerns, for example, the phrasing of observation (i) on p4.*

We are grateful the Reviewer for this useful comment. The title was changed as "Tuning of the flat band and its impact on superconductivity in $\text{Mo}_5\text{Si}_{3-x}\text{P}_x$."

Point (i) in page 4 was reformulated as: "The flat band approaches the Fermi level at $x \gtrsim 1.3$, which leads to a sudden decrease of $\langle v_F \rangle$."

- 3. p2, left-hand column below figure 1: "Figs. 1 (c) and (d)" - this should be (b) and (c).*

Corrected.

- 4. Figure 1(a): missing labels for the different elements in the structure representation.*

The labels for Mo(Si) and P elements are added.

- 5. Figure 1, caption: "The top right panel represent the first Brillouin zone...". As stated elsewhere in the text, this should instead read "The top left panel...".*

Corrected.

- 6. Figure 1(b): The colors used in the legend for fields of respectively 0 T and 9 T are the wrong way round.*

Corrected.

REVIEWERS' COMMENTS

Reviewer #1 (Remarks to the Author):

The authors have well addressed the comments and I now recommend the paper for publication.

Reviewer #2 (Remarks to the Author):

I have worked through the response of the authors to the reports of both referees.

The authors have dealt with the questions and issues raised by both referees in an exhaustive manner.

I believe the manuscript now reflects a unified view among authors and referees, and I recommend that the manuscript may be forwarded for publication.